# Functional Outcome at Short and Middle Term of the Extracorporeal Shockwave Therapy Treatment in Lateral Epicondylitis: A Case-Series Study

**DOI:** 10.3390/jcm9030633

**Published:** 2020-02-27

**Authors:** Gianluca Testa, Andrea Vescio, Stefano Perez, Vincenzo Petrantoni, Giulio Mazzarella, Luciano Costarella, Vito Pavone

**Affiliations:** Department of General Surgery and Medical Surgical Specialties, Section of Orthopedics and Traumatology, A.O.U. Policlinico-Vittorio Emanuele, University of Catania, Via Santa Sofia 78, 95123 Catania, Italy; gianpavel@hotmail.com (G.T.); andreavescio88@gmail.com (A.V.); stefanoperez91@gmail.com (S.P.); vpetrantoni1@gmail.com (V.P.); giulox89.gm@gmail.com (G.M.); lcostarella@yahoo.it (L.C.)

**Keywords:** upper limb, ESWT, conservative treatment, PRTEE

## Abstract

Lateral epicondylitis (LE) of the humerus is a chronic degeneration of wrist extensor tendons at their attachments to the lateral epicondyle of the humerus. There is not a common consensus on a specific therapeutic algorithm, but Extracorporeal Shockwave Therapy (ESWT) is widely used. The purpose of this study is to evaluate the clinical benefits of low dose ESWT in LE-affected patients in short and medium follow-up. Between January 2015 and December 2017, 60 patients (38 male, mean age 52.2 ± 10.1 years, the duration of the disease was 3.6 ± 1.3 months) were clinically evaluated using visual analog scale (VAS) and Patient Rated Tennis Elbow Evaluation Test (PRTEE-I) scores before treatment, at one, three, six and 12 months after treatment. According to the VAS and PRTEE-I scoring systems, all patients achieved an improvement of pain and functional outcome comparing the baseline results with one, six and 12 months values. Low dose ESWT is a safe and effective treatment of LE in the short and middle term. In elderly subjects, patients with a long disease history, or those with occupational and sportive risk factors, a longer persistence of the symptomatology could be observed.

## 1. Introduction

Lateral epicondylitis (LE) of the humerus, also known as “tennis elbow,” was first described in 1873 by Runge as a chronic degeneration of the wrist extensor tendons at their attachments to the lateral epicondyle of the humerus [1]. LE is one of the most common causes of elbow pain and its prevalence is estimated to be between 1% to 3% of the population, especially middle-aged people, with equal gender distribution [2]. Although a definite aetiologic factor cannot be identified in most cases, LE can develop from activities that involve repetitive extension and/or supination of the forearm (tennis, typing, playing musical instruments, manual works). It is for this reason that the most frequently involved muscles are the extensor carpi radialis brevis (ECRB), together with the supinator and other extensor muscles of the forearm [3]. The disturbance was traditionally considered to be a degenerative inflammatory process, but further studies have demonstrated that inflammation is histologically minimal [4]. LE thus has to be considered tendinosis, i.e., a degenerative process caused by recurrent microtraumas and subsequent pathological reparative response [4,5].

The diagnosis of LE is usually based on an accurate clinical evaluation that shows pain and tenderness on the lateral epicondyle, functional impairment and positivity to provocation tests: Mill’s test and Cozen’s test [6]. In addition, LE can be confirmed by musculoskeletal ultrasound [7]. This tendinopathy is actually identified by code FB55.1 in 11th International Classification of Diseases (ICD) Revision [8].

Treatment of LE can be either conservative or surgical in some cases but there is no consensus on a specific therapeutic algorithm. Conservative treatment is employed in the majority of the cases and includes rest, physiotherapy, application of brace or sling, NSAID’s, corticosteroid and Platelet-Rich-Plasma (PRP) injections and Extracorporeal Shockwave Therapy (ESWT) [5].

ESWT has become a widely employed approach in different musculoskeletal pathologies, including the upper limb soft tissues diseases [9], chronic Achilles tendinopathy [10] and plantar fasciitis [11]. Although the therapeutic mechanism of this treatment is controversial, some studies suggested that the shear stress and compression forces induced and maintained by ESWT should stimulate the extracellular matrix binding proteins and endothelial cell homeostasis and the reaction of the bonelacuno-canalicular [12], which should promote the neovascularization, improve angiogenesis, enhance tissue perfusion and reduce the necrotic zone and tissue apoptosis [13,14,15]. However, the efficacy of ESWT for LE is still debated in the literature.

The purpose of this study is to evaluate the clinical benefits and the efficancy of ESWT in LE affected patients in short and medium follow-up. 

## 2. Materials and Methods

### 2.1. Sample

Between January 2015 and December 2017, 107 LE affected patients treated whit ESWT were retrospectively reviewed. In all admitted patients, the following demographic and clinical data were captured: gender, age, involved side, symptoms duration, comorbidities and domiciliary therapy, occupation and sports activities (Table 1).

The inclusion criteria were as follows: (1) confirmed diagnosis of LE with positive at Cozen and Mill test; (2) patient aged between 35–80 years; (3) no previously received shock wave treatment.

The exclusion criteria were as follows: (1) presence of a different or multiple elbow problems, cervical or other upper extremity pathology; (2) history of elbow joint surgery; (3) open or pathological fracture; (4) rupture of the elbow tendon; (5) neurological affection; (6) pregnancy, hemostatic disorder, tumor or local or systemic infection of the upper extremity and implanted pacemaker; (7) follow-up less than 12 months and (8) incomplete treatment protocol.

Before treatment starts, enlightened consent forms were obtained from the patients. All subjects were clinically evaluated before treatment, at one, three, six and twelve months after treatment. After the first four sessions treatment, the patients, who manifested the persisting of the symptomatology, underwent a second cycle of therapy at least 6 months after the first. The cohort was subsequently divided into two groups: Group A, patients who underwent to only one four-session treatment, and Group B, patients who underwent a second four-session treatment.

### 2.2. Treatment Protocol

All the patients received four sessions of focused ESWT with the energy of 0.09 mJ/mm^2^ with a weekly interval using Electromedical device OssaTron equipment. Patients were placed in a sitting position with the superior limb positioned at 90° of elbow flexion, with the palm positioned in the prone position. The trigger points were identified by palpation of the painful areas during the first physical examination and were later balanced through shock waves. After skin protective gel application, a shock wave at a frequency of 4 Hz and a voltage of 14 KeV with 800 shots and pressure of 1.9 bars was performed on the painful point at each ESWT session. The treatment protocol included stretching and eccentric exercises for 50 minutes a day four times a week, use of ice before and after every session. Another four sessions of ESWT treatment was given in positive Cozen and Mill test and visual analog scale (VAS) > 6 subjects at 6 months follow-up.

### 2.3. Clinical Assessment

Clinical and functional outcomes of all patients were evaluated, before treatment, at one, six, and 12 months after treatment, using the VAS [16] score and the Italian version of the Patient Rated Tennis Elbow Evaluation Test (PRTEE-I) score [17]. VAS score is a tool useful to measure the patient-reported pain intensity consisting of a 0 to 10 scale (10 = severe pain; 0 = no pain). PRTEE-I is a specific assessment tool in LE patients consisting of 15 items and three subgroups (pain, special activities, and daily living activities). Higher scores indicate increased pain and functional disability (0 = no disability).

### 2.4. Statistical Analysis

Continuous data are presented as means and standard deviations, as appropriate. The analysis of variance test and Tukey–Kramer method were used to compare the means age, symptoms duration and clinical assessment preoperatively and at one, six, and 12 months postoperatively in the initial cohort and between the two groups. The χ^2^-test was used to verify the homogeneity of the two groups based on gender, and laterality and to compare the Predisposing Occupations and Sports. The selected threshold for statistical significance was *p* < 0.05. All statistical analyses were performed using the 2016 GraphPad Software (GraphPad Inc., San Diego, CA, USA).

## 3. Results

### 3.1. Sample

Of 107 LE affected patients treated with ESWT, 69 were considerate eligible for the study, but five disregarded the treatment protocol and four did not complete the requested follow-up. Finally, 60 patients—38 (63.3%) male and 22 (36.7%) female—were selected for the study. The mean age was 52.2 ± 10.1 years (range 42–63), the duration of the disease was 3.6 ± 1.3 months (range 2–6 months). 67.0% of the cohort performed an ECRB high involving occupation (30.0%) or sports activities (37.0%). After the first four sessions treatment, 12 patients needed an additional therapy cycle and were included in the Group B. Demographic characteristics of the two groups were reported in Table 1. Comparing the mean age (F_1,41_ = 6.68; *p* = 0.01), symptoms duration (F_1,58_ = 15.30; *p* = 0.01), and predisposing occupation and sport (*p* = 0.01) of the two groups showed statistical differences, while the two cohort were similar regarding gender (*p* = 0.35) and laterally (*p* = 0.89). After 12 months, three patients, included in Group 3, were unsatisfied with the ESWT treatments. No complication and adverse reaction were recorded.

### 3.2. Clinical Assessment

#### 3.2.1. Visual Analog Scale (VAS)

According to the VAS scoring system, all patients achieved an improvement of pain (F_3,236_ = 95.40; *p* < 0.00001) comparing the baseline results with values for one (*p* < 0.00001), six (*p* < 0.00001), and 12 months (*p* < 0.00001). Similar data were found comparing the one-month result with the values for 6 (*p* < 0.00001) and 12 months (*p* < 0.00001). No statistical differences were found comparing the VAS score finding between the six months and the last follow-up (*p* = 0.86) (Table 2). Comparing Group A and B, statistically significant differences were found (F_7,232_ = 35.68; *p* < 0.00001); despite the fact that the symptomatology was similar at baseline (Group A vs. B *p* = 0.99), 1 (Group A vs. B *p* = 0.35) and 12 months (Group A vs. B *p* = 0.35), and an improvement of the pain was shown in Group A compared the Group B (*p* < 0.00001).

#### 3.2.2. Patient Rated Tennis Elbow Evaluation Test (PRTEE-I) Score

According to PRTEE-I scoring system, all patients had an improvement of functional outcome (F_3,236_ = 144.99; (*p* < 0.00001) comparing the pre-treatment results and one (*p* < 0.00001), six (*p* < 0.00001) and 12 (*p* < 0.00001) months. The score in each follow-up (*p* < 0.00001), except when comparing PRTEE-I values 6 months after treatment and 12 months after treatment (*p* = 0.73). Similar PRTEE-I values were found comparing the preoperatively (*p* = 0.99), one month (*p* = 0.33), and final follow-up scores (*p* = 0.76) of the two groups. Group A showed better functional outcome after 6 months from the treatment (*p* = 0.0034) (Table 3).

## 4. Discussion

This report showed that ESWT is an effective and efficient conservative treatment in LE affected patients in the short and middle term. Moreover, in younger subjects, for those with a recent disease history, or without occupational and sportive risk factors, a faster recovery was found without the necessity of a second cycle of treatment; after 12 months the outcomes were similar to the patients with additional intervention.

Over 40 different modalities of treatment of lateral epicondylitis, used either alone or in combination, have been reported [18]. Despite the several disadvantages and limitations, such as pregnancy, acute infection, malignant tumor, and coagulopathy [19], as well as a lack of standard treatment protocol, defining the proper number of session per week, number of impulses for each session, type, and energy of shockwave [20], several studies have reported good results of ESWT in decreasing pain, improving the functional outcome [21,22] in chronic patients and in acute presentation. Compared to other common conservative treatment, ESWTs were found superior to ultrasonics therapy and corticosteroid injection [23] in the LE treatment [24].

The reason of the ESWT beneficial action could be detected in the mechanical stimulus provided, which promotes increased pro-inflammatory mediators such as interleukins and metalloproteases [11]. Waugh et al. analyzed the in vivo the biological response to extracorporeal shockwaves, and they showed that in the dialysate tendon, there were some predominant cytokines, some of which had an elevated concentration immediately after ESWT, and then remained significantly elevated for several hours post-ESWT [25].

The use of ESWT in LE was initially controversial due to the study’s contradictory conclusion. In particular, the protocol treatment was wildly debated; in fact, no common consensus is present regarding the use of local anesthesia, the use of various shock waves devices with different application parameters, and the use of NSAIDs. In 2002, Haaken et al., in a multicentric, randomized, blinded, placebo-controlled clinical trial highlighted that low dose ESWTs were not superior when compared to the placebo group for the treatment of chronic lateral epicondylitis in short term, even if local anesthesia-associated [26]. Similar results were found by Chung et al., who confirmed the ineffectiveness of low-energy no-focused shock wave in a 5-week follow-up [27]. Contrarily, in 2003, Melikyan et al. [28] found that the standardization of the treatment protocol and parameters improved the functional outcome and reduced the pain. Subsequently, in 2004, Rompe et al [29], demonstrated the successful results of low dose ESWT, without the application of local anesthesia and the use of pain killers, when a single shock wave device and standardized parameters were used [29]. Little evidence of long-term benefit from ESWT for lateral epicondylitis has been shown by Staples et al. [30]. Recently, in a narrative review, Reilly et al. [31] did not recommend a nerve block or an anesthetic in adduction to ESWT; they concluded that the NSAID’s use can disrupt normal inflammatory pathways that may be responsible for treatment response. Moreover, the authors verified good functional outcomes of ESWTs in LE patients, supporting the theory that previous poor results study studies have created a variety of placebo ESWT conditions, not achieving complete blinding and comparative effectiveness.

Our treatment protocol provided four sessions of focused ESWT with a energy of 0.09 mJ/mm^2^ with a weekly interval. In each session the shock waves were at a frequency of 4 Hz and a voltage of 14 KeV with 800 shots and a pressure of 1.9 bars. No local anesthesia or NSAID’s soministration were performed. The only additional treatment included stretching and eccentric exercises for 50 minutes a day four times a week and the use of ice before and after every session. Similar treatment protocols were described with good results in the literature. 

In a case-control study, Pettrone et al. [32] described successful results of patients managed with ESWT, compared to the placebo group, with a reduction of the VAS score from 74 ± 15.8 baseline to 37.6 ± 28.7 post-treatment after 12 weeks, and at the same follow-up an improvement of functional outcome (*p* < 0.01 respect the baseline). Bayram et al. [6] reported a decrease of pain at the rest, compression and activities 1-month after the treatment compared to baseline (PRTEE pretreatment = 91.50 ± 11.24; PRTEE post-treatment = 55.83 ± 11.69); interestingly Köksa et al. [22] showed a decrease of pain symptomatology while resting, stretching, working, and nighttime pain just after 2 weeks of shockwaves. Speed et al. [33] evaluated the comparison between the treatment of ESWT in patients affected by LE of a least 3 months and a placebo group, but contrary to previous studies, their data showed a 35% successful outcome in the ESWT group and 34% in the placebo group, with no differences between the cohort. Cause of failure of the treatment could be the protocol, which provided a moderate-dose shock wave therapy (1500 impulses at 0.12 mJ/mm^2^) performed once per 3 months [22]; for this reason, the weekly treatment seems the most appropriate. Similarly, Guler et al. [34] reported partial-satisfactory results (improvement in pain and functional outcome, but no significant difference in the grasp and pinching strength compared to the placebo group), but suggested applying a method involving alternative doses and/or different dosage intervals or before surgical treatment in refractory or relapsing cases.

Our findings confirm the efficacy of low dose ESWT to reduce pain and improve the functionality of the LE-affected patients after only 1 month from the treatment and maintain the results at least one year. Taheri et al., in a clinical trial, comparing low dose and high dose focused ESWT, highlighted similar results in pain reduction especially in the first 3 months [35]. Considering the radial ESWT, not significant advantages were found in efficacy in reducing pain or improving function when compared with sham radial ESWT [36]. Moreover, our study identifies some risk factors to the persistence of the symptomatology such as age, duration of disease, and the performing of ECRB high involving occupation and sports activities, which could prejudice the satisfaction of the therapy, making the recovery longer. The literature is sparse in evidence supporting the recurrence protective factors, but dominant hand occurrence, being older, female gender, and a history of smoking are predisposing factors to LE and poor functional outcome [37]. Particularly interesting, for future prospective, is the study of Vitali et al. [38]. The authors highlighted that the combination of ESWT and dietary supplementation produced an increased bioavailability of the supplement to the tendon tissue, due to the neo-angiogenic properties [39,40].

Limitations of this study were: its retrospective nature, the lack of a control group, and the mid-term follow-up. Grip strength and Range of motion assessment, Chair test, Thomas test, and tennis elbow test were performed during the first clinical exam but were not included in the clinical evaluation during the study in order to avoid non-objective measurements.

## 5. Conclusions

Low dose extracorporeal shock wave therapy is a safe and effective treatment of LE in the short and middle term. In elderly subjects, patients with a long disease history, or with occupational and sportive risk factors a longer persistence of the symptomatology could be observed.

## Figures and Tables

**Table 1 jcm-09-00633-t001:** Groups’ demographics. M = male; F = female; R = Right; L = Left.

Group	Patients	Gender		Mean Age (Years)	Symptoms Duration (Months)	Side		Predisposing Occupation	Predisposing Sport
		M	F			R	L		
Sample	60	38	22	52.2 ± 10.1	3.6 ± 1.3	41	19	18	22
Group A	48	29	19	47.9 ± 13.4	2.1 ± 2.7	33	15	13	15
Group B	12	9	3	56.5 ± 8.1	5.2 ± 0.8	8	4	5	7

**Table 2 jcm-09-00633-t002:** VAS score results at preoperatively, one, six, and 12 months after treatment.

Group	Patients	Preoperatively	One Month	Six Months	12 Months
Sample	60	7.5 ± 0.8	5.1 ± 1.3	3.2 ± 2.0	3.0 ± 2.1
Group A	48	7.5 ± 0.8	4.9 ± 1.2	3.0 ± 2.1	2.9 ± 2.2
Group B	12	7.7 ± 0.7	6.2 ± 2.3	6.3 ± 3.3	4.2 ± 2.3

**Table 3 jcm-09-00633-t003:** PRTEE-I score results at preoperatively, one, six, and 12 months after the treatment.

Group	Patients	Preoperatively	One Month	Six Months	12 Months
Sample	60	73.3 ± 8.1	51.8 ± 12.6	33.7 ± 13.4	31.5 ± 12.2
Group A	48	72.9 ± 6.3	48.6 ± 13.5	32.0 ± 8.8	30.3 ± 10.8
Group B	12	75.7 ± 9.4	56.3 ± 9.4	45.3 ± 15.6	35.7 ± 13.6

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
