# Peer review of "Functional Outcome at Short and Middle Term of the Extracorporeal Shockwave Therapy Treatment in Lateral Epicondylitis: A Case-Series Study"

_jcm, 2020, doi:10.3390/jcm9030633_

Round 1

Reviewer 1 Report

The authors present a large case series on outcomes in treatment of lateral epicondylitis using series of shockwave. My major concern is helping to differentiate how this report advances knowledge as larger studies have been conducted with arguably higher LoE in form of 7 RCT. Prior to further consideration of this large case series, the introduction and discussion needs to help synthesize how these current data differ. Clarifying the type of shockwave used (radial vs focused), energy flux density, whether clinical focusing technique was used, and how exclusions were made to create the final sample need to be explained.

Author Response

Reviewer #1

The authors present a large case series on outcomes in treatment of lateral epicondylitis using series of shockwave. My major concern is helping to differentiate how this report advances knowledge as larger studies have been conducted with arguably higher LoE in form of 7 RCT.

  1. A) We appreciate your comments, useful for improving our manuscript.

Q1) Prior to further consideration of this large case series, the introduction and discussion needs to help synthesize how these current data differ.

A1) Useful therapeutic mechanism evidences were added in the introduction, while recent metanalyses and systematic reviews of randomized controlled trials evidences were included in the discussion.

Q2) Clarifying the type of shockwave used (radial vs focused), energy flux density,

A2) The requested information’s were added

Q3) whether clinical focusing technique was used, and how exclusions were made to create the final sample need to be explained.

A3) The inclusion and exclusion criteria were illustrated in methods

Reviewer 2 Report

Dear Authors,

Thank you for the opportunity of reviewing your interesting paper. I have provided my comments and recommendations. Please add your modifications and improvements addressing to my suggestions. In my opinion, our paper is suitable for publication in the Journal of Clinical Medicine after Minor Changes listed below.

General comments:

Lateral epicondylitis (LE) is well known as tennis elbow and it is still one of the most often diagnosed pathology of the upper extremity. It should be emphasized that the incidence of LE is 1% / 1000 patients per year and its prevalence is 1–3% of adult patients per year. There is a need to search for novel, effective and safe modalities for LE management and shock wave therapy (ESWT) might be a beneficial one. It should be pointed out that a lot of papers in this subject have been previously published and the novelty is quite limited. PubMed search reveals 376 records for “shock wave and lateral epicondylitis” keywords. However, as the Authors noticed, the clinical utility and effectiveness of ESWT for patients with LE is still controversial in the worldwide literature. Therefore, they designed a case-series study to assess the clinical benefits and the efficacy of ESWT in patients affected with LE considering both, short and medium follow-up observations.

Specific comments:

  1. Introduction: In general, it’s informative and contains a background introducing the reader into the subject. However, can’t agree with your statement (line 49) that “the therapeutic mechanism of this treatment is not fully understood yet”. Please see more details about the mechanism ESWT action in example papers listed below. Please also explain more details about this issue and add these three relevant papers A-C in your references section. A: Romeo P, Lavanga V, Pagani D, Sansone V. Extracorporeal shock wave therapy in musculoskeletal disorders: a review. Med Princ Pract. 2014;23(1):7–13; B: Dymarek R, Halski T, Ptaszkowski K, Slupska L, Rosinczuk J, Taradaj J. Extracorporeal shock wave therapy as an adjunct wound treatment: a systematic review of the literature. Ostomy Wound Manage. 2014;60(7):26–39; and C: Wang CJ. Extracorporeal shockwave therapy in musculoskeletal disorders. J Orthop Surg Res. 2012;7:11.
  2. Introduction: Please be consistent in using any abbreviations in the entire paper (EL, ESWT, VAS, etc.). Explain to them when using the first time and then use only abbreviations!
  3. Materials and Methods / Sample: Why have you used a retrospective protocol and case series study instead of a prospective and clinical controlled design? Please add this into your limitations. Has your study been registered in the Clinical Registry Platform, even retrospectively? If not, why?
  4. Materials and Methods / Sample: Have the patients undertaking pharmacological therapy or continued their rehabilitation program been excluded from the study?
  5. Materials and Methods / Sample: I don’t understand this sentence, it seems to be a kind of error, please to correct (lines 74-77): “3. Results This section may be divided by subheadings. It should provide a concise and precise description of the experimental results, their interpretation as well as the experimental conclusions that can be 77 drawn.”
  6. Materials and Methods / Treatment Protocol: Please specify if you used a radial or focused device, it’s really important from the technical and research point of view.
  7. Materials and Methods / Treatment Protocol: Did you use a coupling ultrasonic gel as a contact medium? Have you noticed and report any ESWT-related adverse effects (pain, bruisers, etc.)? Please explain if ESWT was delivered only to the tendons, or also to the muscles or to the musculotendinous junction?
  8. Materials and Methods / Clinical Assessment: Assessment methods are also limited. Why didn’t you use such typical tests for LE as grip strength, ROM, Chair test, Thomas test, and tennis elbow test? Please comment on this in your limitations.
  9. Results: Please add in the text the reference for Table 3.
  10. Discussion: You tried to explain (lines 143-145) that “… in younger subjects, or with a recent disease history, or without occupational and sportive risk factors a faster recovery was found without the necessity of the second cycle of treatment, …” There are any other potential factors affecting these positive outcomes? How about using medications or some other types of treatments, physiotherapy, exercises? Please explain.
  11. Discussion: Why there is no discussion and citation to previously published metanalyses and systematic reviews of randomized controlled trials in terms of ESWT and LE?

Despite the above-mentioned remarks, the paper is interesting and has a scientific value. In my opinion, this paper should be recommended for publication in the Journal of Clinical Medicine after Minor Changes.

Best regards, Reviewer.

Author Response

Reviewer #2

Dear Authors,

Thank you for the opportunity of reviewing your interesting paper. I have provided my comments and recommendations. Please add your modifications and improvements addressing to my suggestions. In my opinion, our paper is suitable for publication in the Journal of Clinical Medicine after Minor Changes listed below.

General comments:

Lateral epicondylitis (LE) is well known as tennis elbow and it is still one of the most often diagnosed pathology of the upper extremity. It should be emphasized that the incidence of LE is 1% / 1000 patients per year and its prevalence is 1–3% of adult patients per year. There is a need to search for novel, effective and safe modalities for LE management and shock wave therapy (ESWT) might be a beneficial one. It should be pointed out that a lot of papers in this subject have been previously published and the novelty is quite limited. PubMed search reveals 376 records for “shock wave and lateral epicondylitis” keywords. However, as the Authors noticed, the clinical utility and effectiveness of ESWT for patients with LE is still controversial in the worldwide literature. Therefore, they designed a case-series study to assess the clinical benefits and the efficacy of ESWT in patients affected with LE considering both, short and medium follow-up observations.

  1. A) We appreciate your comments, useful for improving our manuscript.

Specific comments:

Q1) Introduction: In general, it’s informative and contains a background introducing the reader into the subject. However, can’t agree with your statement (line 49) that “the therapeutic mechanism of this treatment is not fully understood yet”. Please see more details about the mechanism ESWT action in example papers listed below. Please also explain more details about this issue and add these three relevant papers A-C in your references section.                                                                                         A: Romeo P, Lavanga V, Pagani D, Sansone V. Extracorporeal shock wave therapy in musculoskeletal disorders: a review. Med Princ Pract. 2014;23(1):7–13;                                                 B: Dymarek R, Halski T, Ptaszkowski K, Slupska L, Rosinczuk J, Taradaj J. Extracorporeal shock wave therapy as an adjunct wound treatment: a systematic review of the literature. Ostomy Wound Manage. 2014;60(7):26–39;                                                                                                           and C: Wang CJ. Extracorporeal shockwave therapy in musculoskeletal disorders. J Orthop Surg Res. 2012;7:11.

A1) the section was clarified and the references added.

Q2) Introduction: Please be consistent in using any abbreviations in the entire paper (EL, ESWT, VAS, etc.). Explain to them when using the first time and then use only abbreviations!

A1) the requested modifies were made.

Q3) Materials and Methods / Sample: Why have you used a retrospective protocol and case series study instead of a prospective and clinical controlled design? Please add this into your limitations. Has your study been registered in the Clinical Registry Platform, even retrospectively? If not, why?

A3) Retrospective study was chosen for easer and faster data collection and the possibility to assess a large-population cohort. The study was not registered to a Clinical Registry Platform.

Q4) Materials and Methods / Sample: Have the patients undertaking pharmacological therapy or continued their rehabilitation program been excluded from the study?

A4) During the ESWT treatment was asked to the patients to perform stretching and eccentric exercises for 50 min a day four times a week, use of ice before and after every session, the patient did not treat with other additional rehabilitation program. No note of the pharmacological therapy was recorded during the treatment

Q5) Materials and Methods / Sample: I don’t understand this sentence, it seems to be a kind of error, please to correct (lines 74-77): “3. Results This section may be divided by subheadings. It should provide a concise and precise description of the experimental results, their interpretation as well as the experimental conclusions that can be 77 drawn.”

A5) The typing error was eliminated

Q6) Materials and Methods / Treatment Protocol: Please specify if you used a radial or focused device, it’s really important from the technical and research point of view.

A6) the requested information was added.

Q7) Materials and Methods / Treatment Protocol: Did you use a coupling ultrasonic gel as a contact medium? Have you noticed and report any ESWT-related adverse effects (pain, bruisers, etc.)? Please explain if ESWT was delivered only to the tendons, or also to the muscles or to the musculotendinous junction?

A7) As reported in Treatment protocol, skin protective gel was applied to avoid bruiscers or other adverse effects. The ESWTs were perform the musculotendinous junction (considered the trigger point)  

Q8) Materials and Methods / Clinical Assessment: Assessment methods are also limited. Why didn’t you use such typical tests for LE as grip strength, ROM, Chair test, Thomas test, and tennis elbow test? Please comment on this in your limitations.

A8) The requested changes were added

Q9) Results: Please add in the text the reference for Table 3.

A9) The reference was added.

Q10) Discussion: You tried to explain (lines 143-145) that “… in younger subjects, or with a recent disease history, or without occupational and sportive risk factors a faster recovery was found without the necessity of the second cycle of treatment, …” There are any other potential factors affecting these positive outcomes? How about using medications or some other types of treatments, physiotherapy, exercises? Please explain.

A10) Young age, recent disease history and the absence of occupational and sportive risk factors were the functional outcome positive predictors. No other variables were examined. The paucity of evidences in literature does not allow to individuate other protective factors for the recurrence and relapse of LE. The use of medications or some other types of treatments, physiotherapy, exercises (except for those included in treatment protocol) were not noted.

Q11) Discussion: Why there is no discussion and citation to previously published metanalyses and systematic reviews of randomized controlled trials in terms of ESWT and LE?

A11) Recent metanalyses and systematic reviews of randomized controlled trials evidences were included in the discussion.

Despite the above-mentioned remarks, the paper is interesting and has a scientific value. In my opinion, this paper should be recommended for publication in the Journal of Clinical Medicine after Minor Changes.

Best regards, Reviewer.

Round 2

Reviewer 1 Report

The authors have made improvements to the manuscript. However, I am still concerned that the full list of studies including RCT on lateral epicondylitis have not been included including Haake 2002, Melikyan 2003, Rompe 2004, Chung 2004, Staples 2008. These studies need to be discussed in detail to help explain differences in the current protocol with mixed findings in studies with higher level of evidence based on study design. A discussion on differences in study design is included in review by Reilly, et al. PM&R Journal 2018

Author Response

Q1) The authors have made improvements to the manuscript. However, I am still concerned that the full list of studies including RCT on lateral epicondylitis has not been included including Haake 2002, Melikyan 2003, Rompe 2004, Chung 2004, Staples 2008.

These studies need to be discussed in detail to help explain differences in the current protocol with mixed findings in studies with a higher level of evidence-based study design.

A discussion on differences in study design is included in the review by Reilly, et al. PM&R Journal 2018.

A1) We appreciate your comments. Every study you cited was discussed and included in the manuscript.

Sincerely yours,
VP
